# Fire Risk Probability Mapping Using Machine Learning Tools and Multi-Criteria Decision Analysis in the GIS Environment: A Case Study in the National Park Forest Dadia-Lefkimi-Soufli, Greece

Yannis Maniatis [1,*], Athanasios Doganis [2] and Minas Chatzigeorgiadis [3]

1 Department of Digital Systems, University of Piraeus, 18534 Piraeus, Greece
2 Terra Mapping the Globe S.A., 15561 Xolargos, Greece; thanos@terra.gr
3 Department of Digital Systems, Graduate Research Assistant at University of Piraeus, 18534 Piraeus, Greece; minas.ch32@gmail.com
* Correspondence: maniatis@unipi.gr

**Abstract:** Fire risk will increase in the upcoming years due to climate change. In this context, GIS analysis for fire risk mapping is an important tool to identify high risk areas and allocate resources. In the present study, we aimed to create a fire risk estimation model that incorporates recent land cover changes, along with other important risk factors. As a study area, we selected Dadia-Lefkimi-Soufli National Forest Park and the surrounding area since it is one of the most important protected areas in Greece. The area selected for the case study is a typical Mediterranean landscape. As a result, the outcome model is generic and can be applied to other areas. In order to incorporate land cover changes in our model, we used a support vector machine (SVM) algorithm to classify a satellite image captured in September 2021 and an image of the same period two years ago to obtain comparable results. Next, two fire risk maps were created with a combination of land cover and six other factors, using the analytic hierarchy process (AHP) on a GIS platform. The results of our model revealed noticeable clusters of extreme high risk areas, while the overall fire risk in the National Park Forest of Dadia-Lefkimi-Soufli was classified as high. The wildfires of 1st October 2020 and 9th July 2021 confirmed our model and contributed to quantification of their impact on fire risk due to land cover change.

**Keywords:** wildfire; fire risk; model; MCDA; AHP; Natura; protected zones; GIS; SVM; land cover change

## 1. Introduction

The frequency of forest fires is rapidly increasing in southern Europe, posing major challenges for Greece, Italy, Portugal, Spain, and France [1]. Wildfires can represent a serious threat to human health and infrastructure, as well as ecosystems and biodiversity [2]. More specifically, the impact of wildfires on human health can be either direct, causing severe physical damage due to burns, or indirect, since the exposure to pollutants such as ozone and PM [3] can lead to serious disorders. In addition, large wildfires can damage properties or critical infrastructure, such as electricity grids and houses, resulting in major economic losses [4]. Finally, wildfires play an important role in ecological balance, in which humans are a part of. The increase in fire frequency in the past few years enhances forest degradation and biodiversity loss [5].

In terms of biodiversity, Greece is one of the richest countries in Europe, having the highest number of flora species among the Balkan countries. In fact, it contains over 5700 different species of flora, 20% of which are endemic to the country. Most of these species are located in the northern regions of the country, thanks to the ideal geographic and climate conditions [6,7]. It seems that species richness of Greece combined with the high

risk of fire in the Mediterranean region—because of its hot and dry summers—makes Greece extremely susceptible to wildfires. Consequently, the average annual burned area caused by wildfires has shown an increasing trend in the past decades. During 2001–2017, wildfires burned an average of 55,000 ha per year, most of which was covered by forested areas [8]. Recently, one of the most disastrous wildfires took place in August 2021 in Evia island, where 34,893.5 ha of forest and 1111.6 ha containing houses and infrastructure were destroyed or seriously affected by fire [9]. As a result, it is critical to study how various factors influence the probability of fire occurrence, in order to create a fire risk layer for the Fire Management Geographic Information

Fire risk expresses the likelihood of a fire occurring during a specific time period and place. The risk is the result of the different hazardous parameters interacting with the conditions of vulnerability, which are present in the region [10]. On the one hand, hazardous parameters describe the danger of fire occurrence and on the other hand, vulnerability expresses the predilection of an area to be negatively affected by wildfire [11]. It is very common for the terms 'fire risk' and 'fire danger' to have interchangeable meanings. The factors that influence the ignition and development of fires constitute the fire danger. Fire ignition can derive from natural causes (mostly thunder), or it can be a result of human activity [12]. According to a study, approximately 93% of fires in Northern Europe are caused by humans, either intentionally or unintentionally [13], and thus the location of populated places and roadways is critical in identifying areas at high risk of fire. The development of fires is influenced by topography, meteorological conditions, fuel condition, and fuel availability [14]. Many studies have shown that vegetation and topography are the key elements responsible for fire severity in many types of forests. [15–18]. Since topographic features influence the distribution of local climate, topography is an important factor in fire propagation. Fires spread quickly across steep and upward slopes, but slowly in places with a downhill slope [19]. Moreover, the probability of fire occurrence may vary in different elevations on the basis of factors such as temperature and vegetation [20]. The topographic wetness index (TWI) [21] is another parameter that contributes to fire spread and ignition. To a certain extent, TWI simulates the impact of topography to soil and fuel moisture [22,23].

Geographic information systems (GIS) is mature technology and effective platform to analyze, visualize, and disseminate spatial and temporal data and information. GIS is a multidisciplinary approach that can combine methods from science, engineering, and the economy with the experience of field officers to produce robust knowledge in firefighting. GIS, besides its analytical capabilities, is the ideal platform for coordination, information exchange, and awareness provision for all involved stakeholders (all levels of authorities, fire department, police, forest services, agricultural coops, citizens, etc.).

Multiple levels of spatial and nonspatial data and information related to fire risk, such as meteorological data, land cover, vegetation features, and topography, in the form of historical information is combined and evaluated to create detailed fire risk maps [24–26]. The information to be used by the fire risk model has to be reliable, the most recent, easy to obtain, and processable with reasonable H/W and S/W resources in order to produce an update fire risk map. The key for the analysis is the determination and assignment of the proper weights between all these pieces of information. Many studies, in particular, have used multi-criteria decision analysis (MCDA) in conjunction with the analytic hierarchy process (AHP), which assigns weights to the influencing parameters, so as to successfully develop fire risk maps [14,24–28]. In the AHP framework, a decision is broken down into a hierarchy of criteria or alternatives, and subsequently one can evaluate the significance of each criterion to the final decision, given the relevant weights between each pair of criteria [29]. After an exhaustive review of the bibliography, we proposed seven factors to be used as criteria in AHP to estimate fire risk: the land cover (LC), the elevation, the aspect, the slope, the TWI, and the distance from roads (DfR) and settlements (DfS). Out of those seven factors, studies have shown that land cover is the most important for estimating fire risk [24,27], especially in cases where the land cover indicates the type of vegetation

that covers the area [28,30]. It is evident that having a detailed and updated depiction of land cover is critical for estimating fire risk. In view of this fact, GIS can be used to classify land cover and vegetation from satellite imagery with the implementation of machine learning algorithms [31,32]. Various algorithms have been used by different studies for land classification, such as k-means clustering [33], maximum likelihood classification [34], and support-vector machines (SVM) [35]. In fact, support-vector machine models have been used for land classification with promising results [31,35].

In this paper, the SVM algorithm was applied to satellite images obtained in 2019 and 2021 in order to create detailed land cover maps for the Natura 2000 (GR1110005) zone, which includes the Dadia-Lefkimi-Soufli National Park Forest in the county of Evros. Subsequently, the use of these land cover maps in combination with six other important fire risk factors can determine the fire risk of the National Park for September 2019 and September 2021. On the basis of the fire risk maps of 2019 and 2021 in conjunction with the burned areas from the past fires of 1st October 2020 and 9th July 2021, we evaluated how substantial land cover changes can affect fire risk mapping. In order to make a valid comparison, we chose to include in the fire risk model factors that remain relatively constant for long periods of time. This creates a baseline fire risk map of our study area.

## 2. Study Area

Our study area is the Natura 2000 zone with codename GR1110005, which coincides with the National Park Forest of Dadia-Lefkimi-Soufli. The study area is located in Evros county, as shown in Figure 1, and it extends from 26.03° E to 26.32° E and from 40.98° N to 41.26° N, covering a total area of 42,481 ha. The climate in the area is Mediterranean, with daytime maximum average temperatures of 32 °C in August and lowest average temperature of 8 °C in January [36]. The average number of rainy days per year is 13.3 and the average yearly rainfall is 732 mm [37]. The lowest point of the study area has a height of 10 m, and the highest point is located in Kapsalo at 620 m [37,38].

The National Park Forest of Dadia-Lefkimi-Soufli contains two protected zones, A1 and A2, which cover an area of 7350 ha. Oak and pine trees make up the majority of the forested areas. The spatial distribution of the different types of trees in the National Park can be divided into two areas. The center is covered with pine trees, whereas the north and southwest are covered mostly with oak trees [39]. During 2019 and 2021, two major fires occurred within the National Park, both of which took place in the southern region, on the northern part of the village Lefkimi. The first burstfire occurred on 1st October 2020, burning approximately 694 ha, and the second one took place on 9th July 2021, burning approximately 242 ha.

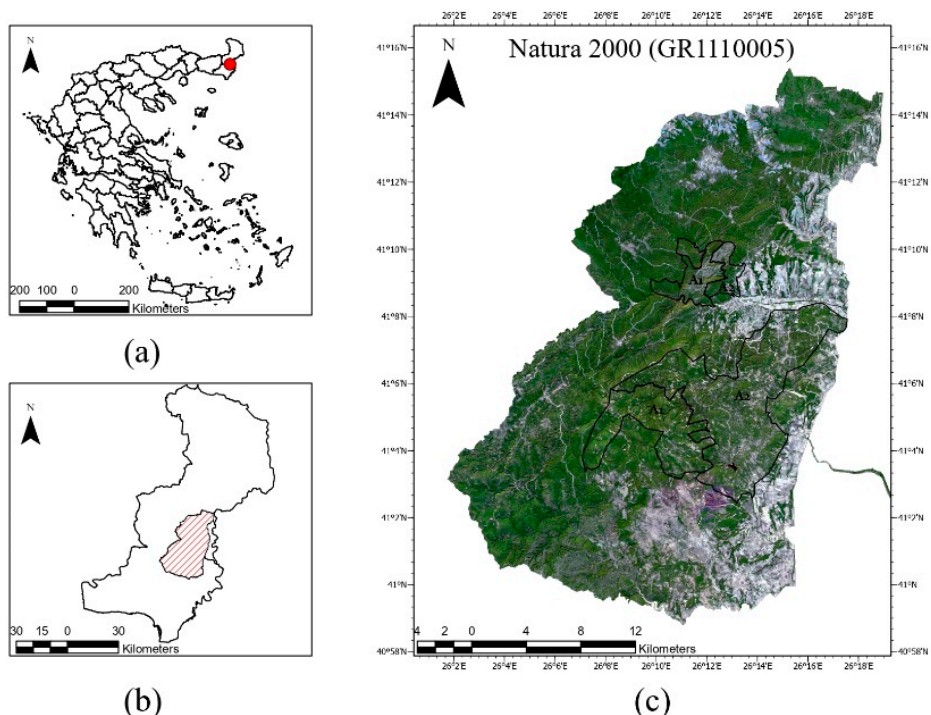

**Figure 1.** (**a**) Location of the study area in Greece (**b**) and in Evros county. (**c**) Longitude and latitude of Natura 2000 area (GR1110005) that coincides with the National Park Forest of Dadia-Lefkimi-Soufli, including the protected zones A1 and A2.

### 3. Data and Methods

To construct the land cover maps, Sentinel-2 images were used, which were captured on 18th September 2019 at 09:06 a.m. and on 27th September 2021 at 09:06 a.m., with minimal cloud coverage (<0.1%) [40]. Both images were level 2 Sentinel-2 products, and thus they had already received atmospheric correction. Next, the spectral bands B03, B04, and B08 were extracted from the original images, with spatial resolution of 10 m, in order to produce the color infrared images needed for the land classification. Finally, to classify the color infrared image, we used a supervised machine learning (SML) model, using the application of the support vector machine (SVM) algorithm. SVM algorithms have been proven to be a reliable method of creating land cover maps from Sentinel-2 images [31,32,35].

For the calculation of the topographic factors, the freely available digital elevation model (DEM) of the Copernicus Land Monitoring Service was used. The Copernicus DEM offers spatial resolution of $25 \times 25$ m, with vertical accuracy of $\pm 7$ m (RMSE). For the purpose of our study, the $1000 \times 1000$ km tile with codename E50N20 was used, and the elevation of our study area was isolated from it [38]. The remaining topographic factors (slope, aspect, and TWI) were derived by analyzing the DEM. The roads and settlement locations were downloaded from open data sources [41,42]. For the raster analysis and calculations mentioned above, the GIS software ArcGIS Pro 2.9.1. was used.

In order to validate our results and examine the impact of the fire on land cover and subsequently the fire risk, detailed burn scar maps from the National Management Body of Dadia-Lefkimi-Soufli Forest National Park were used [37], which depict the extent of fires that occurred during October 2020 [43] and July 2021 [44]. VIIRS hotspot locations were also used to validate the spatial extend of the burned areas [45]. The workflow of our method is represented in Figure 2.

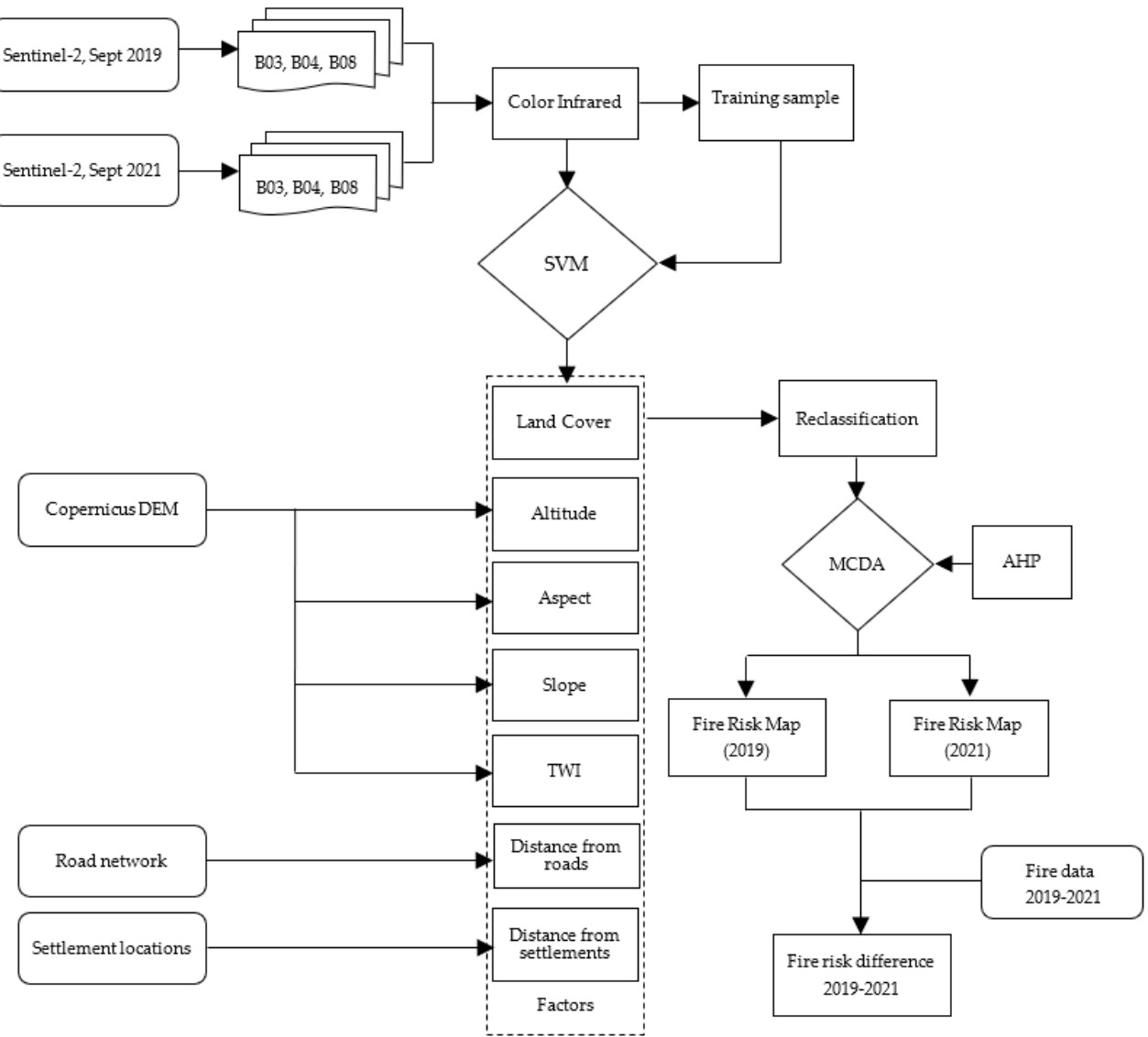

**Figure 2.** Flowchart of the applied methodology.

*3.1. Factors*

3.1.1. Land Cover (LC)

To create the land cover maps, we used an SVM algorithm to process color infrared images from September 2021 and September 2019, which were derived from the combination of Sentinel-2 B03, B04, and B08 bands. The resulting color infrared images are presented in Figure 3, in which vegetation appears in shades of red, bare land in cyan or white, and water in black.

Color infrared images can help distinguish among different plant types, depending on their leaf characteristics [46]. Inside the National Park of Dadia-Lefkimi-Soufli, oak and pine trees account for more than the 70% of vegetation [39]. Oak trees belong to the broad-leaved tree family, and therefore they appear in brighter red in the color infrared image. In Figure 3, oak trees can be seen as clusters of bright red in the northern and southwestern parts of the national park. Pine trees have thinner leaves, and thus they appear to have a darker red color. Finally, shrubs and low grass appear in faint red.

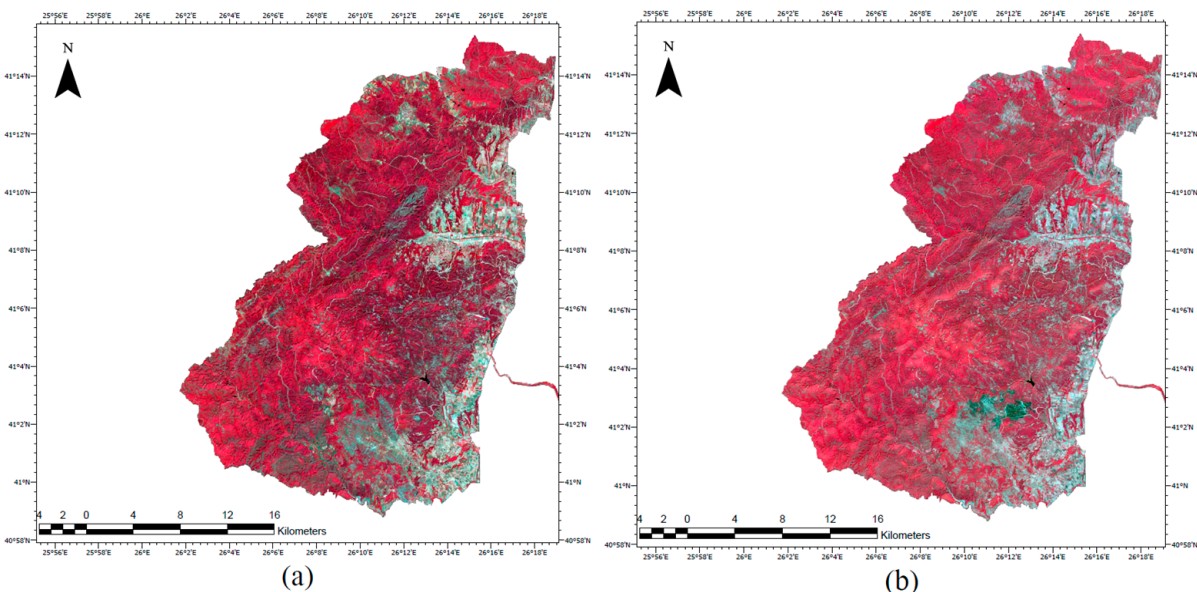

**Figure 3.** Color infrared image of the study area produced by the combination of spectral bands B03, B04, and B08 (**a**) from September 2019 and (**b**) from September 2021.

To classify the images, we considered six classes: pine forest, oak forest, shrubs and low grass, bare land, water bodies, and built-up areas. To train the SVM algorithm, we carefully gathered multiple samples of homogeneous parts from each image, representing one of the six classes. After the classification of the image, we made some adjustments to the product image, mainly to distinguish some parts of bare land from the buildup areas. The accuracy of the land cover classification was estimated using the Kappa coefficient, which was found to be 0.87. The final land cover maps from September 2019 and September 2021 of the National Park of Dadia-Lefkimi-Soufli are presented in Figure 4a,b, respectively.

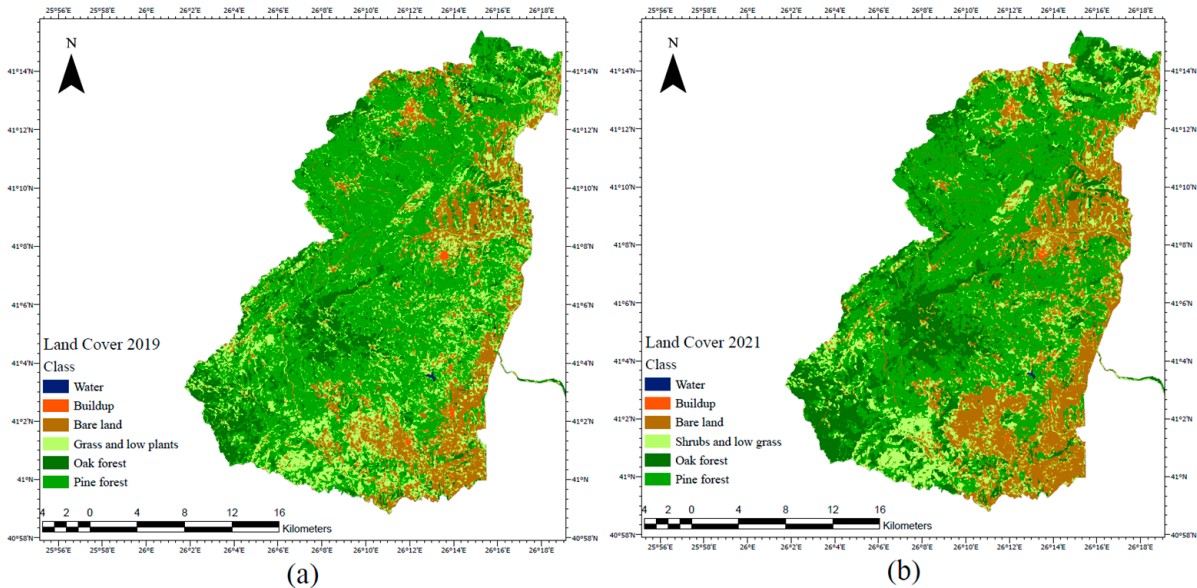

**Figure 4.** Final land cover map produced by the SMV algorithm (**a**) from September 2019 and (**b**) from September 2021.

Each type of tree has different flammability properties. Considering that pine trees are more flammable than oak trees, the forest areas were classified accordingly [47]. Finally, since water bodies cannot ignite, they were classified with the fire risk class 'no risk'.

The classification of land cover, based on the fire risk, is shown in Table 1, and the final reclassified risk map of the land cover maps is presented in Figure 5.

**Table 1.** Fire risk classification of land cover.

| Land Cover Class | Risk Class | Risk Description |
|---|---|---|
| Pine forest | 5 | Extremely high |
| Oak forest | 4 | High |
| Shrubs and low grass | 3 | Medium |
| Bare land | 2 | Low |
| Buildup | 1 | Extremely low |
| Water body | 0 | No risk |

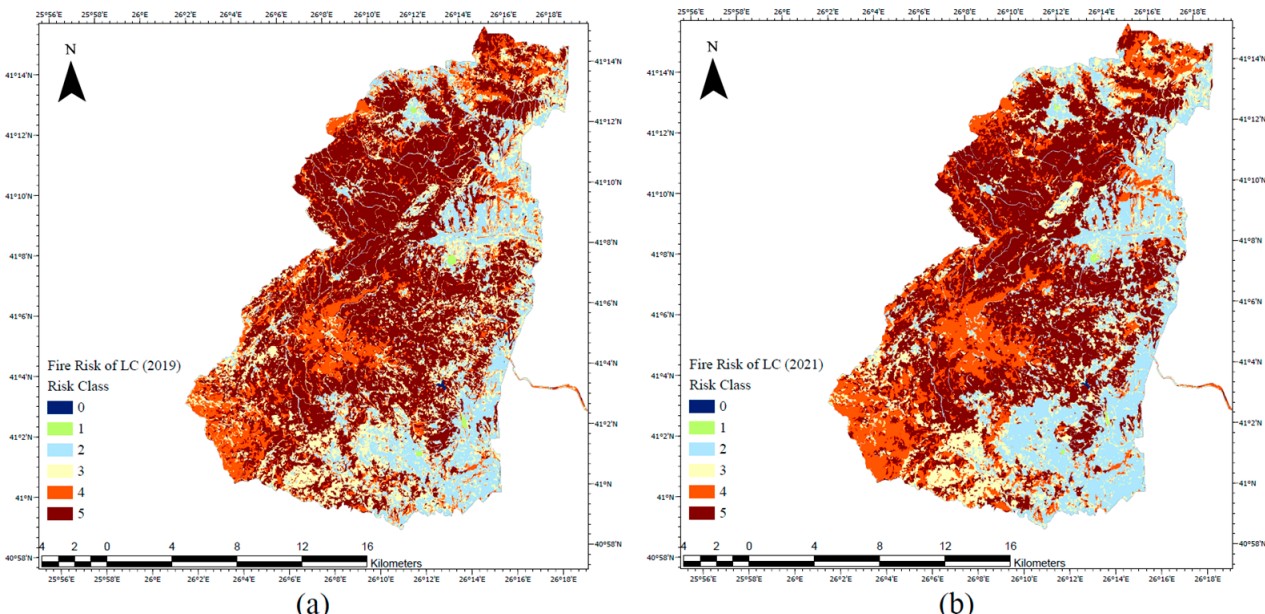

(a)                                                      (b)

**Figure 5.** The fire risk classification of land cover (**a**) for September 2019 and (**b**) for September 2021.

### 3.1.2. Altitude

Altitude influences the humidity of vegetation and temperature. Vegetation in high altitudes has higher rates of humidity and lower temperature [31]. Moreover, high altitudes usually have lower vegetation density. Considering the topographic characteristics of the area, we distributed the fire risk into five classes, as shown in Table 2. Elevation in our study area, according to DEM [38], ranged from 10 m meters to 645 m meters. The altitude raster and the final reclassification of fire risk appear in Figure 6.

**Table 2.** Fire risk classification of altitude.

| Altitude (m) | Risk Class | Risk Description |
|---|---|---|
| 10–100 | 5 | Extremely high |
| 100–200 | 4 | High |
| 200–300 | 3 | Medium |
| 300–400 | 2 | Low |
| >400 | 1 | Extremely low |

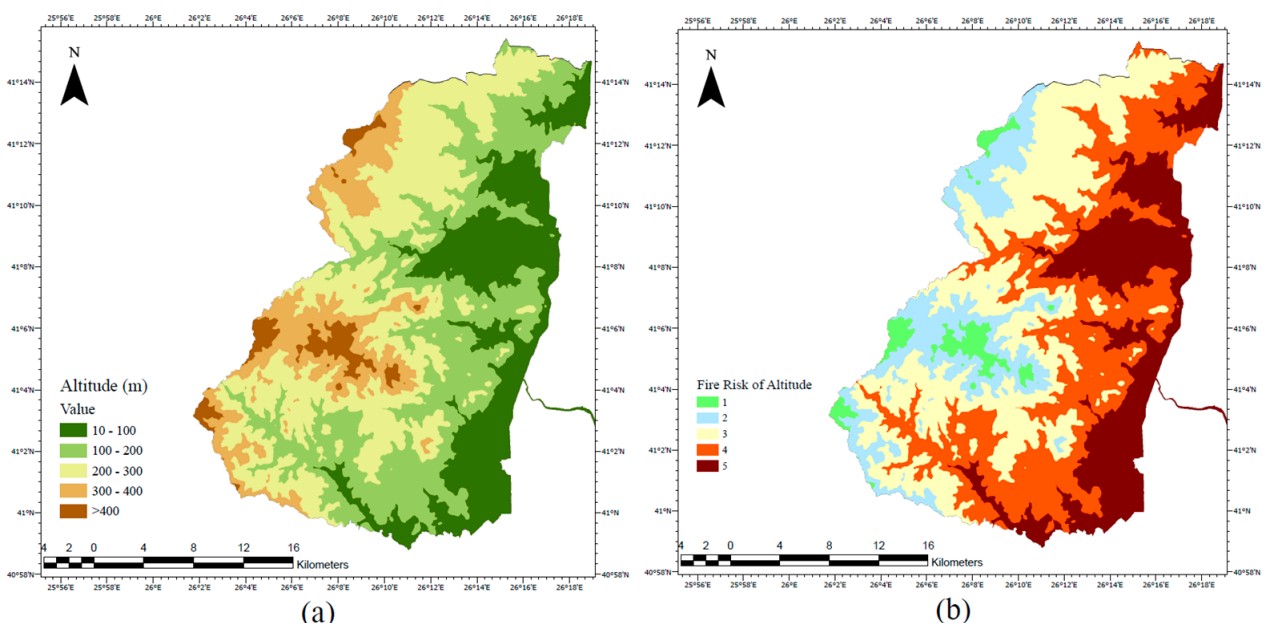

**Figure 6.** (**a**) Altitude. (**b**) Fire risk classification of altitude.

### 3.1.3. Aspect

In the northern hemisphere, south-oriented slopes receive more sunlight, and thus the vegetation loses humidity faster and becomes more flammable [28]. Moreover, because of the difference in sunlight distribution among the different orientations of slope, the southern aspects usually have more dense vegetation. Less humidity and dense vegetation results in higher fire risk, and thus vegetation facing south is more flammable. The fire risk classification of the aspect appears in Table 3 [25,28].

**Table 3.** Fire risk classification of aspect.

| Aspect | Risk Class | Risk Description |
|---|---|---|
| South | 5 | Extremely high |
| Southeast–East | 4 | High |
| Northeast | 3 | Medium |
| North | 2 | Low |
| Flat–Southwest–West–Northwest | 1 | Extremely low |

The aspect derives from the DEM and the is presented with different colors depending on the orientations. The final aspect raster along with reclassified fire risk map can be seen in Figure 7.

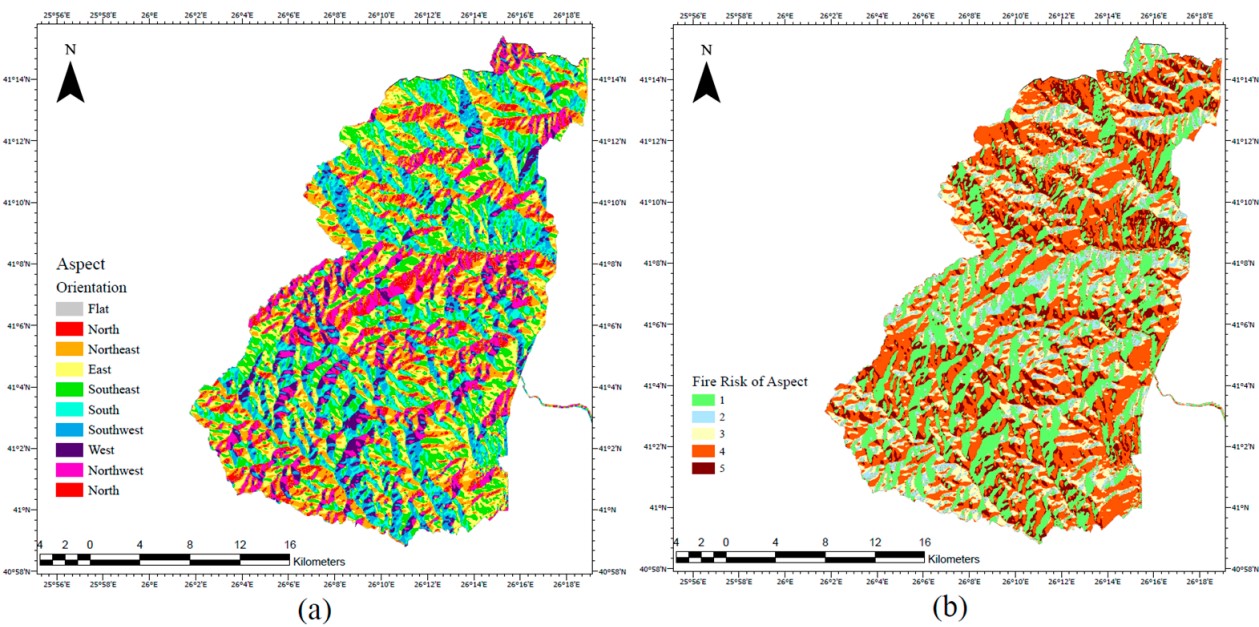

**Figure 7.** (**a**) Aspect. (**b**) Fire risk classification of aspect.

### 3.1.4. Slope

Fire propagates faster on steeper slopes because the flames can reach higher vegetation more easily at great surface angles [30]. Moreover, on steep slopes, water runoff increases, resulting in less soil moisture [24]. Both of these factors make areas with steeper slopes have a higher risk of fire. We derived the fire risk classification of slope as shown in Table 4 [25]. In order to calculate the slope raster, we used the DEM and chose to present the results in percentage. The slope raster of the National Park Forest of Dadia-Lefkimi-Soufli along with the fire risk map of the slope is presented in Figure 8.

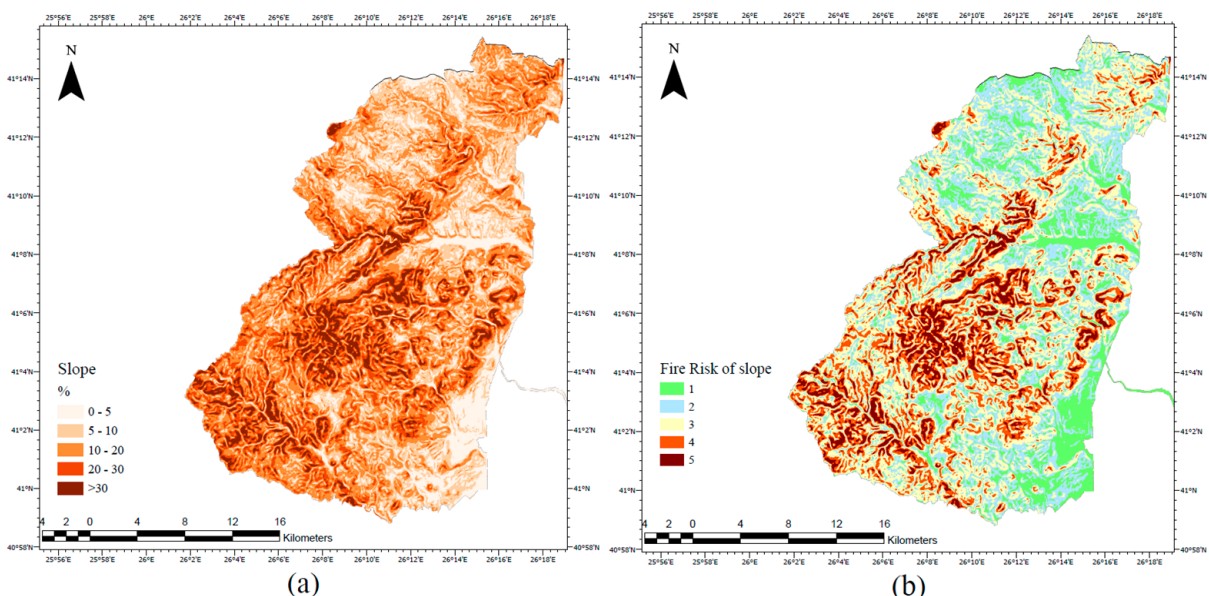

**Figure 8.** (**a**) Slope. (**b**) Fire risk classification of slope.

**Table 4.** Fire risk classification of slope.

| Slope (%) | Risk Class | Risk Description |
|---|---|---|
| >30 | 5 | Extremely high |
| 20–30 | 4 | High |
| 10–20 | 3 | Medium |
| 5–10 | 2 | Low |
| 0–5 | 1 | Extremely low |

### 3.1.5. Topographic Wetness Index (TWI)

The TWI can simulate water concentration can be derived from topography. The presence of water affects soil moisture and makes the surrounding vegetation harder to ignite [22]. We calculated the TWI of the study area from the total catchment area, the flow width, and slope from the DEM [21]. The risk classification of the TWI is presented in Table 5. The TWI raster of the study area along with the fire risk map of TWI is shown in Figure 9.

**Table 5.** Fire risk classification of TWI.

| TWI | Risk Class | Risk Description |
|---|---|---|
| 4–6 | 5 | Extremely high |
| 6–7 | 4 | High |
| 7–8 | 3 | Medium |
| 8–9 | 2 | Low |
| >9 | 1 | Extremely low |

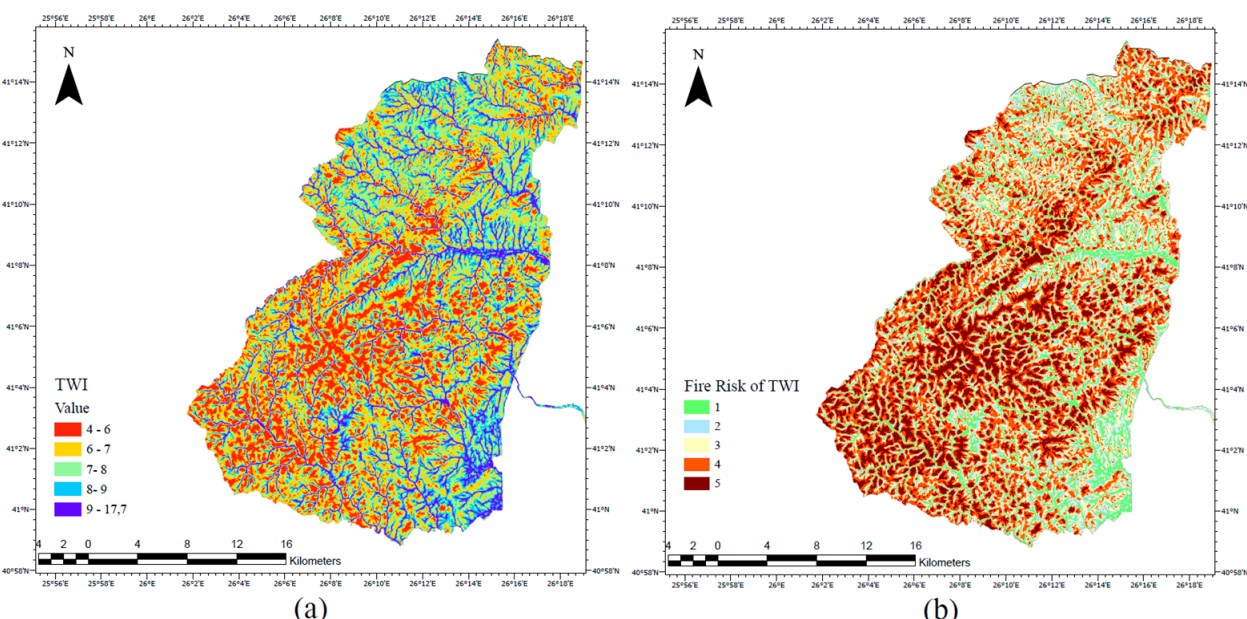

**Figure 9.** (**a**) TWI. (**b**) Fire risk classification of TWI.

### 3.1.6. Distance from Roads

Human activities near roads can be the cause of fire ignition, and therefore the areas surrounding the road network are at a higher fire risk [48]. To attribute fire risk to those areas, we took into consideration previous studies along with the structure of the road network [24,28]. The first 200 m near the road network was determined to be at high risk of fire, and afterwards the risk decreased by one class at 200 m intervals. The classification of fire risk is presented in Table 6. Multi-buffer rings of 200 m each were calculated around

each road segment and transformed to rasters in order to be incorporated into the model (Figure 10).

**Table 6.** Fire risk classification of the area around the road network.

| DfR (m) | Risk Class | Risk Description |
| --- | --- | --- |
| 0–200 | 5 | Extremely high |
| 200–400 | 4 | High |
| 400–600 | 3 | Medium |
| 600–800 | 2 | Low |
| >800 | 1 | Extremely low |

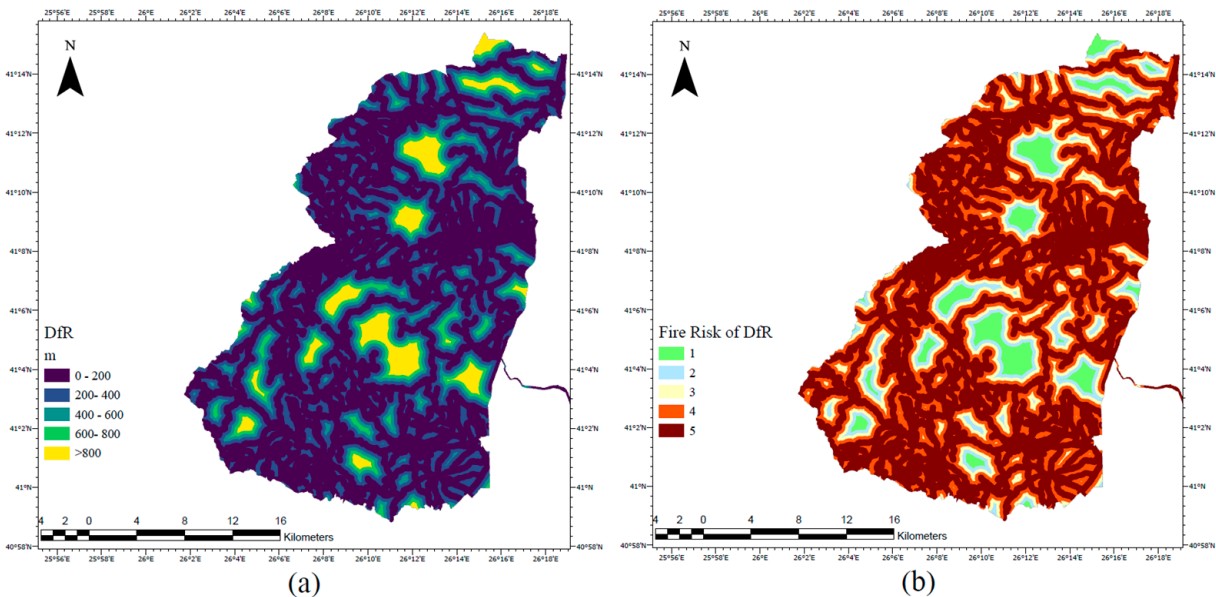

(a)      (b)

**Figure 10.** (**a**) Rasterized buffer zones every 200 m from the road network. (**b**) Fire risk classification based on the road network.

### 3.1.7. Distance from Settlements

The distance around settlement locations affects the risk of fire similarly to that of the road network. The areas closer to settlements are in higher risk than those farther away [13]. To distribute the fire risk, we took the spatial extent of the settlements into consideration. Since most of the settlements inside and near our study are small, we estimated that their average extent is 500 m. Taking this into account, we assigned the area inside a radius of 900 m around the settlements to be in extreme risk of fire, and afterwards the risk decreased by one class at 400 m intervals, as it is depicted in Table 7. Buffer zones using the aforementioned distances were applied around each settlement. The results and the assigned fire risk is shown in Figure 11.

**Table 7.** Fire risk classification of the area around settlement locations.

| DfS (m) | Risk Class | Risk Description |
| --- | --- | --- |
| 0–900 | 5 | Extremely high |
| 900–1300 | 4 | High |
| 1300–1700 | 3 | Medium |
| 1700–2100 | 2 | Low |
| >2100 | 1 | Extremely low |

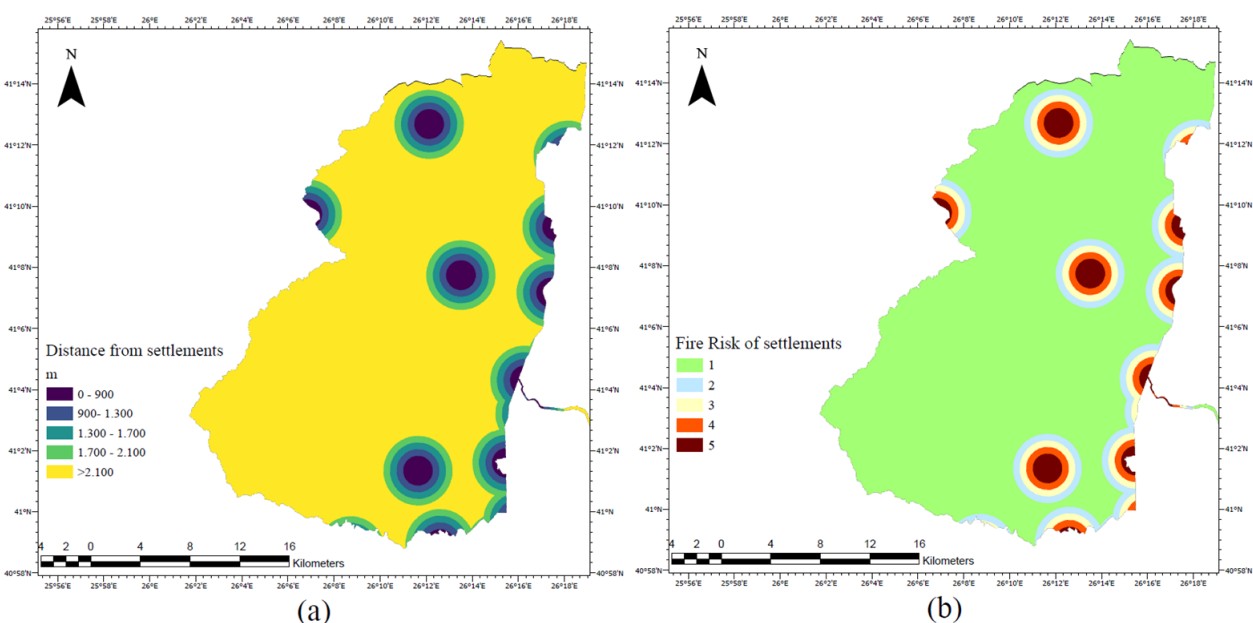

**Figure 11.** (**a**) Rasterized buffer zones of the distance from settlements. (**b**) Fire risk classification of the area around settlement locations.

### 3.2. Attribution of Weight to the Factors

To calculate the weight of each factor, we used the AHP as a multi-criteria method. The AHP can estimate the significance of each factor given the pairwise comparisons among each one of the seven factors [29]. In our model, each factor belongs to one major category that influences fire risk. Particularly, slope, DEM, aspect, and TWI reflect the impact of topography and to some extent fuel condition on fire risk. The distance from roads and settlements captures the impact of human activity on fire ignition. Finally, land cover depicts the alternations of fire risk, due to fuel availability and fuel type. In order to estimate the pairwise comparisons, we consulted past studies and expert opinions. We considered the land cover to be the most important factor for this estimation [24,25,27]. Moreover, human activity plays a key role in fire risk identification, since in most cases, humans are the main cause of fires [13]. Therefore, the distance from roads and settlements has a serious impact on fire risk. The most important topographic factor is TWI, since it has a direct correlation with soil humidity, whereas the rest topographic factors contribute less to the overall fire risk. The final distribution of pairwise comparisons is presented in Table 8. Afterwards, the weights were calculated using the mathematical procedure established by Thomas L. Saaty [49]. The final weights of each factor are also shown in Table 8.

**Table 8.** Pairwise comparisons of fire risk factors along with the assigned weight.

|  | Land Cover | Altitude | Aspect | Slope | TWI | DfR | DfS | Weight |
|---|---|---|---|---|---|---|---|---|
| Land cover | 1 | 3 | 3 | 3 | 3 | 2 | 2 | 0.27 |
| Altitude | 0.33 | 1 | 3 | 2 | 0.5 | 0.33 | 0.33 | 0.09 |
| Aspect | 0.33 | 0.33 | 1 | 0.5 | 0.25 | 0.33 | 0.33 | 0.05 |
| Slope | 0.33 | 0.5 | 2 | 1 | 0.5 | 0.33 | 0.33 | 0.07 |
| TWI | 0.33 | 2 | 4 | 2 | 1 | 0.33 | 0.33 | 0.12 |
| DfR | 0.5 | 3 | 3 | 3 | 3 | 1 | 3 | 0.23 |
| DfS | 0.5 | 3 | 3 | 3 | 3 | 0.33 | 1 | 0.17 |
| SUM | 3.32 | 12.83 | 19 | 14.5 | 11.25 | 4.65 | 7.32 | 1 |

To verify the consistency of our comparison estimations, we calculated the consistency ratio (CR) by applying the following equations [30],

$$CI = (\lambda max - n)/(n - 1) \tag{1}$$

$$CR = CI/RI \tag{2}$$

The λmax in Equation (1) is the perturbated eigenvalue of the matrix constructed by the pairwise comparisons, as depicted in Table 8. n is the order of the matrix, n = 7. The consistency index (CI) in Equation (1) measures the difference between λmax and the exact eigenvalue, n. The CR in Equation (2) is calculated from the random consistency index (RI) [50], as shown in Table 9.

**Table 9.** Values of the random consistency index (RI).

| n | 1 | 2 | 3 | 4 | 5 | 6 | 7 |
|---|---|---|---|---|---|---|---|
| RI | 0 | 0 | 0.58 | 0.9 | 1.12 | 1.24 | 1.32 |

According to Table 9, RI = 1.32 for seven factors. Subsequently, concerning our pairwise estimations, CR = 0.07. Since CR < 0.1, the estimations of the pairwise matrix were consistent.

The fire risk maps were calculated by the weighted sum of all factors

$$\text{Fire Risk} = 0.27 * LC + 0.09 * Altitude + 0.05 * Aspect + 0.07 * Slope + 0.12 * TWI + 0.23 * DfR + 0.17 * DfS \tag{3}$$

## 4. Results

The fire risk maps for September 2019 and September 2021, with spatial resolution of 25 m × 25 m, were calculated using Equation (3), and they are presented in Figure 12, along with the relative fire risk scale value. The risk class for most areas was unchanged, which is expected. The parameters used in our model remained relatively invariant for long periods of time. Therefore our risk map represents the baseline fire risk in the area [24].

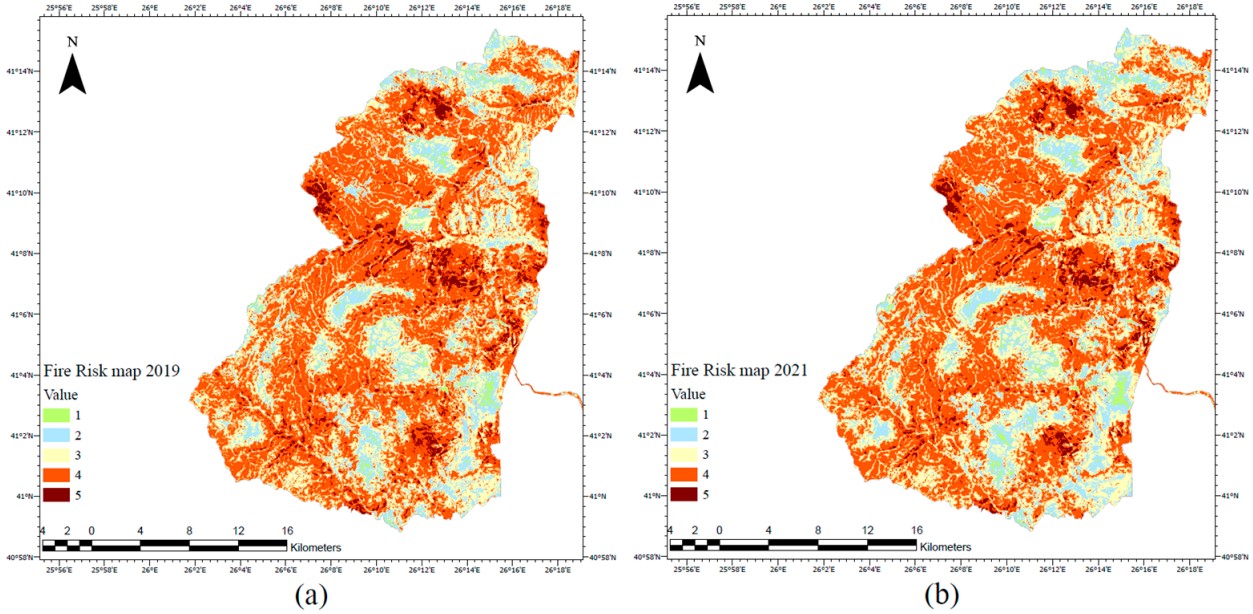

**Figure 12.** Fire risk in the National Park Forest of Dadia-Lefkimi-Soufli (**a**) for September 2019 and (**b**) for September 2021.

The average fire risk in the National Park Forest of Dadia-Lefkimi-Soufli was high for both years. Specifically, 50% of the total 42,481 ha of our study area for 2019 was considered to be at high fire risk. Similarly, 48% of the total area was considered to be at high fire risk for 2021. Moreover, for both years, 5% of the National Park was at extremely high risk of fire.

The extreme fire risk areas form clusters. The most noticeable ones are located near the center of the National Park, spreading along the line that connects the points with coordinates 41°10' N, 26°7' E and 41°6' N, 26°16' E. Two additional extremely high risk areas were detected. The first one is located in the northwest of the National Park, near the settlement of Giannouli, and the second one is located in the southeast of the park, near the settlement of Lefkimi. The overall distribution of fire risk for September 2019 and September 2021 in the National Park is presented in Table 10.

**Table 10.** The distribution of fire risk in the National Park of Dadia-Lefkimi-Soufli for September 2019 and September 2021.

| Risk Class | Risk Description | Fire Risk Areas (Sept 2019) | Fire Risk Areas (Sept 2021) |
|---|---|---|---|
| 5 | Extremely high | 5% | 5% |
| 4 | High | 50% | 48% |
| 3 | Medium | 33% | 34% |
| 2 | Low | 11% | 12% |
| 1 | Extremely low | 1% | 1% |

Within the National Park Forest of Dadia-Lefkimi-Soufli, two major fire incidents took place between September 2019 and September 2021. The first one was recorded in October 2020, and the second one in July 2021 [46]. In order to validate our results and measure the impact of change in the land cover on the fire risk, due to the fires, each fire incident was examined separately as follows.

*4.1. Impact of Fire in October 2020*

The fire of October 2020, as stated by the fire department, started in the north of the village of Lefkimi, near the southwest extreme high fire risk area, and burned approximately 694 ha. According to VIIRS hotspot measurements, the brightness temperature during the fire ranged from 24 °C to 81 °C [46]. Moreover, the area affected by the fire before the fire occurrence was considered at high risk. In fact, according to our fire risk map of 2019, 41% of the area was classified as high risk and 36% as medium risk. The fire extent and the fire risk map of 2019, along with the distribution of the fire risk inside the affected area, are presented in Figure 13.

The fire had a significant impact on the fire risk in the area. The change in land cover we identified with the SVM algorithm passed on the fire risk and was captured by the difference among the fire risk maps inside the extent of the affected area before and after the fire incident. The fire risk of the area dropped from high to medium–low risk. The high-risk areas dropped from 41% before the fire to 10% after the fire, while the low-risk areas increased by 17%. These changes are attributed to the loss of vegetation from the fire on 5 October 2020. The fire extent related to the fire risk map of 2021, along with the updated distribution of the fire risk inside the affected area, is presented in Figure 14.

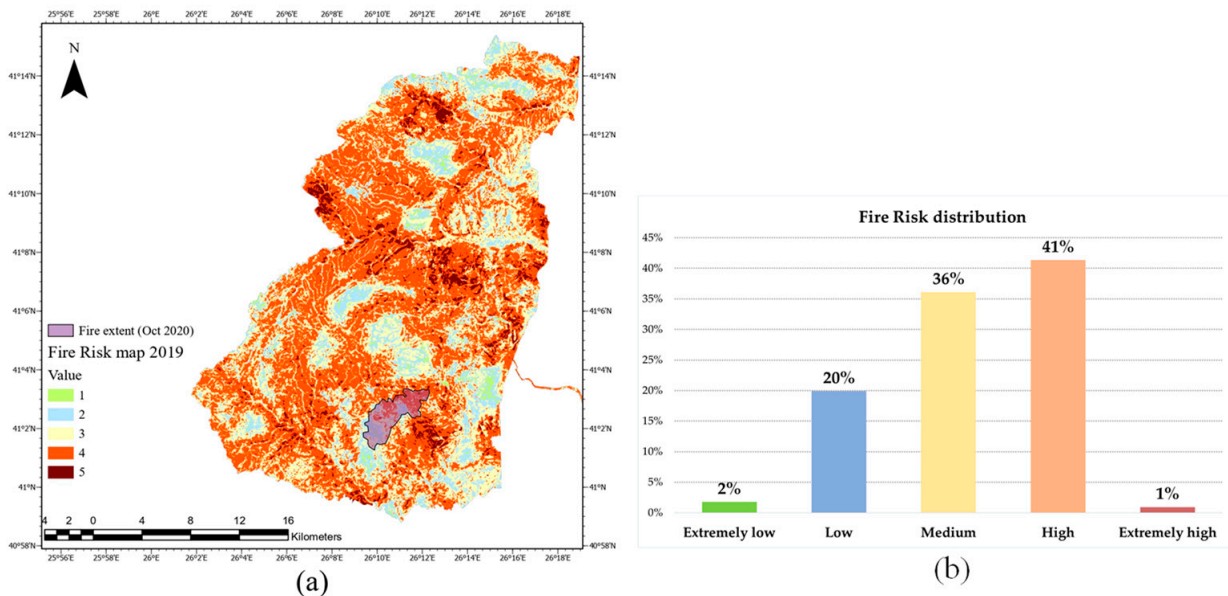

**Figure 13.** (**a**) The extent of the fire on 5 October 2020 relative to the fire risk map of 2019. (**b**) The fire risk distribution inside the affected area before the fire on 5 October 2020.

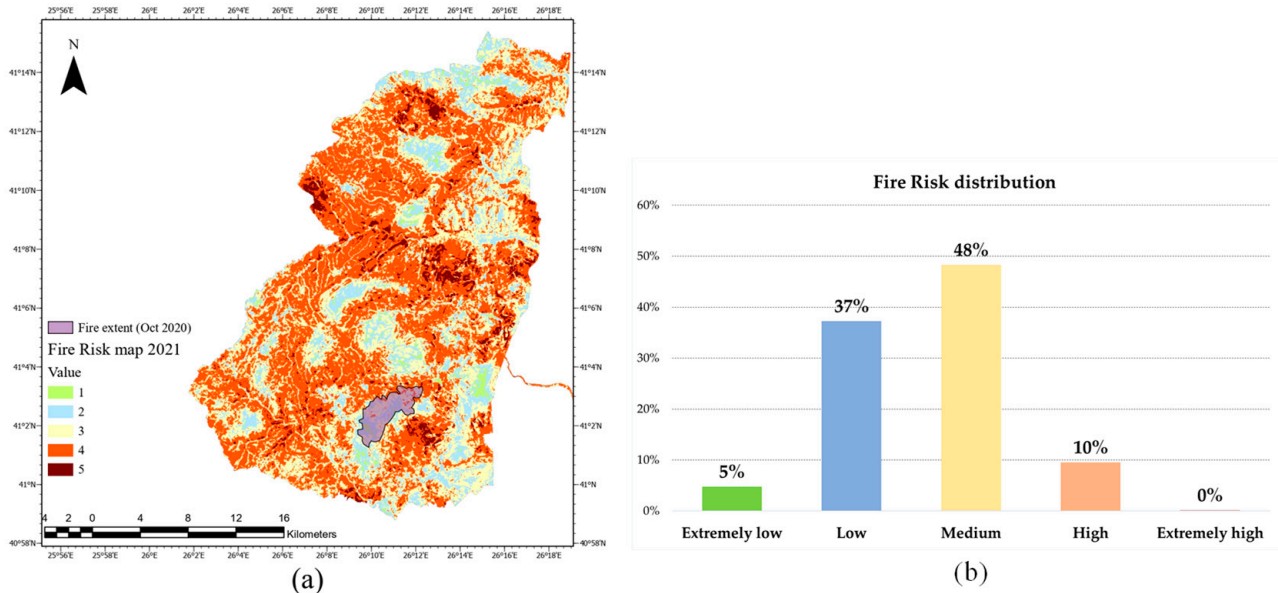

**Figure 14.** (**a**) The extent of the fire on 5 October 2020 relative to the fire risk map of 2021. (**b**) The updated fire risk distribution inside the affected area, after the fire on 5 October 2020.

*4.2. Impact of Fire in July 2021*

The second serious fire incident in the National Park Forest of Dadia-Lefkimi-Soufli was recorded on 9th July 2021, in the north of the settlement of Lefkimi. The fire burned approximately 242 ha of forested areas, and it was close to the limits of the fire that occurred in October 2020, as is shown in Figures 14 and 15. According to VIIRS hotspot measurements, the brightness temperature during the fire ranged from 59 °C to 32 °C [46]. According to our model, the affected area before the fire occurrence was considered to be at high risk of fire. In particular, 59% of the overall area was classified as having a high fire risk, and 10% as having an extremely high fire risk. On the contrary, only 4% of the area

was classified as low and 0% as extremely low fire risk. The detailed distribution of the fire risk inside the affected area (according to the risk map of 2019) is presented in Figure 15.

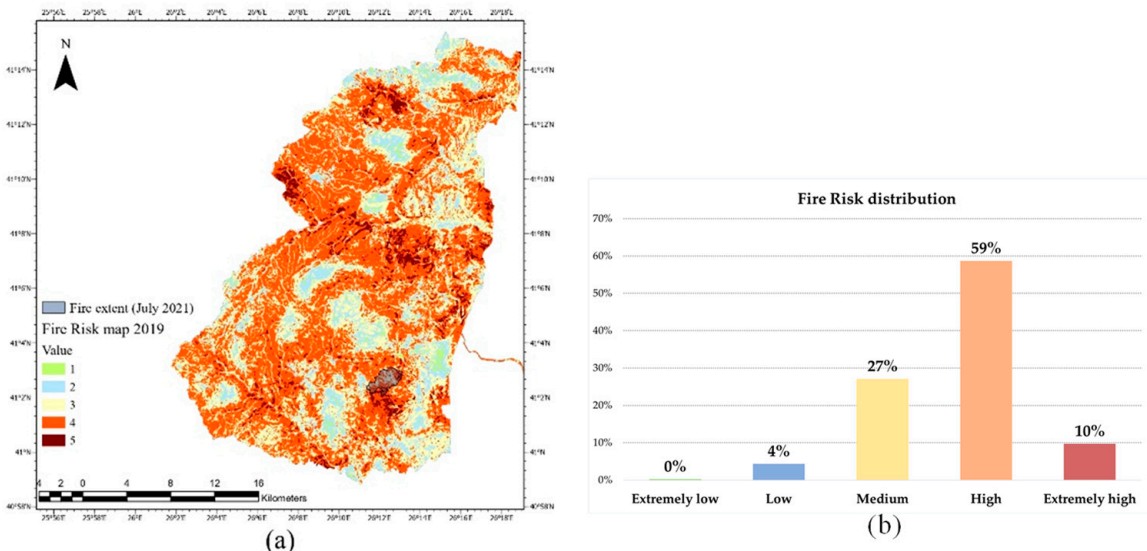

**Figure 15.** (**a**) The extent of the fire on 9 July 2021 relative to the fire risk map of September 2019. (**b**) The fire risk distribution inside the affected area, before the fire on 9 July 2021.

The fire had a significant impact on the overall fire risk classification of the area. The fire risk of the affected area after the fire was classified as medium with 49%. After the fire incident, the high-risk areas dropped from 59% of the whole area to 26%, while the extremely high risk areas dropped from 10% to 3%. On the other hand, areas categorized as low risk increased by 18%. The fire extent related to the fire risk map of September 2021, along with the updated distribution of the fire risk inside the affected area, is presented in Figure 16.

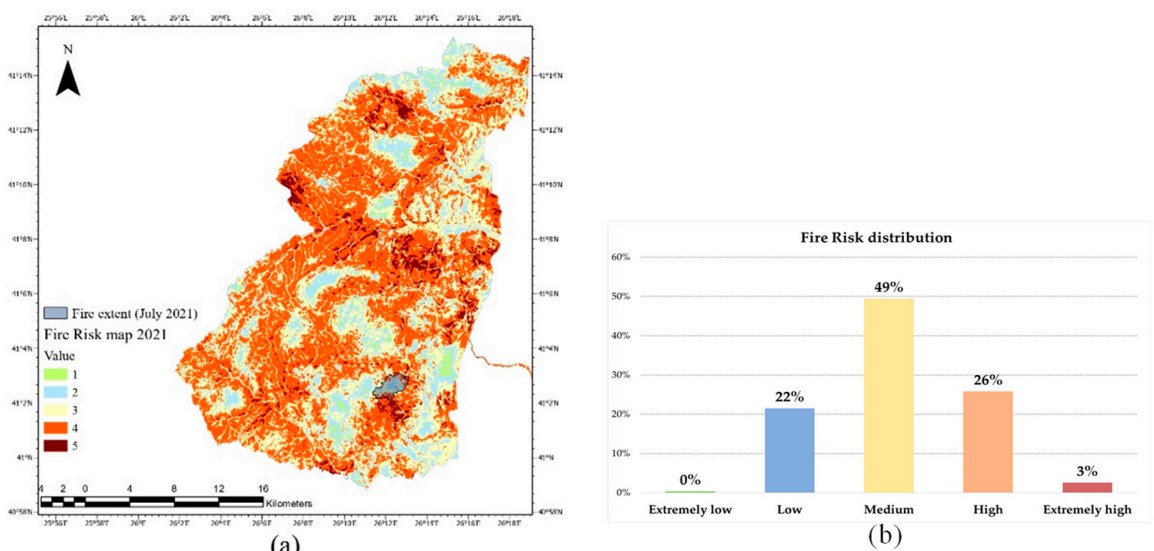

**Figure 16.** (**a**) The extent of the fire on 9 July 2021 relative to the fire risk map of 2021. (**b**) The updated fire risk distribution inside the affected area, after the fire on 9 July 2021.

## 5. Discussion

In this study, we combined elevation, slope, aspect, TWI, land cover, settlement location, and road networks to create a fire risk model. Subsequently, we applied the model

to the National Park Forest of Dadia-Lefkimi-Soufli. Topography, vegetation, and human activity are the major categories represented in our model that affect wildfire generation and spread. Similar factors have been used by other studies, in combination with AHP to assign weights, in order to calculate fire risk in various areas [14,24–28,30]. In our analysis, land cover was proved to be the factor with the highest weight. This does not mean that topography and human activities are of less importance. In fact, human activity is the main cause of wildfire ignition. Approximately 93% of fires in Northern Europe are caused by humans, either intentionally or unintentionally [13]. Land cover, besides having the highest weight in the resulting model, also changes more frequently compared to human activities (road network, settlements) and topography. For this reason, we derived land cover classification from Sentinel 2 imagery since they are georeferenced, frequently updated, freely available, and have suitable spatial resolution. Classification was carried out with the use of SVM algorithm, since it has already been used in similar applications with promising results [31,35].

With the combination of AHP (to determine the fire risk) and the SVM algorithm (to classify the land cover), we managed to identify the baseline fire risk of the National Park Forest of Dadia-Lefkimi-Soufli for September 2019 and September 2021. According to those fire risk maps, most of the areas in the National Park Forest of Dadia-Lefkimi-Soufli are classified as high risk. More specifically, the map of 2021 reveals that 5% out of the total area of the National Park was classified as extremely high risk, and 48% was high risk. Consequently, fire risk distribution in the National Park suggests that local authorities should be at high alert, especially during heatwaves and near the areas classified as extreme high risk.

We also examined in detail the impact of the land cover change on fire risk in the areas affected by the two major fire incidents (October 2020 and July 2021). It is concluded that the average risk of those areas dropped significantly, while the rest of the fire risk map remained relatively unchanged between September 2019 and September 2021. It is evident that land cover changes caused by past fires have a significant drop on the fire risk of the affected areas. The main cause is the loss of the highly flammable pine tree forest near the settlement Lefkimi. Considering the effect of land cover changes on fire risk mapping, it is important that fire risk management plans incorporate those changes and reallocate resources accordingly on a local scale. In this way the SVM algorithm, along with other classification algorithms [33–35,51], can offer a powerful tool for updating fire risk maps year by year. Finally, it is notable to point out that both October 2020 and July 2021 fires started near areas classified as 'extreme high' fire risk. This is yet another validity indicator of the proposed fire risk model. Additionally, the above-mentioned fire incidents spread mostly at areas classified as 'high risk' (Figures 13 and 15).

We acknowledge that daily risk maps at the national [52] and European levels [53], which are mostly derived from weather data, are freely available and easily accessible to all stakeholders, including the citizens. Prediction models based on weather information alert civil protection and fire services on areas of increased readiness. Fire risk models and maps (such as the one proposed in this study) are supplementary to weather data, indicating high risk areas usually at higher spatial resolution and where proactive measures can, or should, be taken. Those measures can include the optimal allocation of observatories and/or fire service areas by using GIS tools such as visibility, network, or other suitable analysis.

We utilized GIS technology for fire risk model development, not only because of GIS's analytical and presentation capabilities, but also due to information dissemination and its integration ability along with other organizational workflows. Developing the proposed model, a key issue addressed is data availability. As the reader can see all data utilized are from reliable [38,40] and freely referenced or downloaded sources [41,42]. As a result, local authorities, which are responsible in specifying precautional policies and measures (especially in high fire risk periods), such as increased supervision, temporal road closures to traffic, prohibition of certain activities, and more optimal resource allocation, have the means to identify areas of higher risk.

The spatial resolution of the presented model is limited to the resolution (25 by 25 m) of the DEM used to calculate it [38]. Higher-resolution DEMs are calculated by mapping agencies during orthophoto production workflows. They can also be acquired by drones [54]. A question of further research and investigation is if higher resolution DEMs help in a better understanding and mapping fire risk or just add higher frequency data, which sometimes are noise rather than actual information. A constrain of our model is the lack of meteorological factors. Furthermore, it is important to note that in order to determine the fire risk of an area, meteorological factors such as temperature, wind, and humidity play a very important role [26]. These factors change dynamically, and therefore it is hard to establish a baseline risk map to compare results from different years. Human activity layers of information (road network and settlements) were derived from OpenStreetMap in testing our model. More important than the accuracy of road network is completeness and the level of update. State agencies and local authorities are advised to use the most updated information they have access to. Other data sources of human activities, such as electricity grids and landfills, which are of great importance, can be incorporated in our model, creating multi-buffer rings, similar to how road network and settlement areas were treated.

## 6. Conclusions

Wildfires, unfortunately, are an inevitable consequence of climate crisis. Every year, we are witnessing more and more devastating wildfires in the western United States, Amazon basin, South Europe, Siberia, Australia, and elsewhere, with a priceless impact on our environment. Understanding and modelling the phenomenon can make us more effective in addressing it. Especially in firefighting, the timely response is the most crucial factor to fight it. The role of fire risk models in conjunction with other GIS analysis tools can provide us useful information for optimal arrangement of all available resources before the ignition of the phenomenon such as selecting supervision locations for areas characterized as 'high risk' and even allocate firefighting trucks for a more immediate response in case of an incident. Knowledge of 'high risk' areas can assist all levels of administration to increase citizen awareness and take targeted proactive measures.

Models and all data needed to support it should be free and easily accessible for agencies and authorities to integrate it with their systems. All necessary data to implement the model can be easily found at Copernicus services and other European or National spatial data infrastructures. The most essential data to classify land cover, Sentinel 2 images, are available freely worldwide through the same services. We will keep working and testing the model in other areas to test its portability. Findings and suggestions will help us to improve it. All future improvements will be embedded and published.

**Author Contributions:** Conceptualization, A.D. and Y.M.; methodology, A.D. and Y.M.; software, A.D.; validation, M.C.; formal analysis, A.D.; investigation, A.D. and M.C.; resources, M.C. and Y.M.; data curation, M.C.; writing—original draft preparation, M.C.; writing—review and editing, A.D. and Y.M.; visualization, M.C.; supervision, A.D. and Y.M.; project administration, A.D. and Y.M.; funding acquisition, Y.M. All authors have read and agreed to the published version of the manuscript.

**Funding:** This research received no external funding.

**Institutional Review Board Statement:** Not applicable.

**Informed Consent Statement:** Not applicable.

**Data Availability Statement:** The 25 m resolution DEM data can be downloaded from Copernicus Land Monitoring Service (https://land.copernicus.eu/imagery-in-situ/eu-dem/eu-dem-v1.1, accessed on 5 October 2021). The Sentilel-2 images for land cover classification are found at Copernicus Open Access Hub (https://scihub.copernicus.eu/dhus/#/home, accessed on 10 December 2021).

**Acknowledgments:** We would like to thank Paraschi Eirini for copy editing the final manuscript.

**Conflicts of Interest:** The authors declare no conflict of interest.

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
