# Peer review of "Fire Risk Probability Mapping Using Machine Learning Tools and Multi-Criteria Decision Analysis in the GIS Environment: A Case Study in the National Park Forest Dadia-Lefkimi-Soufli, Greece"

_applsci, doi:10.3390/app12062938_

Round 1

Reviewer 1 Report

The article is very interesting in term of predicting the fire hazard in a national park. However, in Table 1 , water consider factor was 1, which I think must be -1 due the water nature. Hence the manuscript will produce a different outcomes. Also, in the discuss there is misleading statement " Although
those maps are extremely useful in fire risk management, they offer very poor spatial resolution, so various fire risk models, like the one we developed, can indicate the detailed
distribution of the fire risk" this must be amended.

I would like to see some temperature values mentioned during those fire incidents to give the reader an idea what risk at what temperature exist.

Please a conclusion consisting from few lines

Reviewer 2 Report

  1. In Section 2,the authors mentioned that “ The climate in the area is Mediterranean, with daytime maximum average temperatures of 32 ° C in August and lowest average temperature of 8 ° C in January. The average number of rainy days per year is 13.3 and the average rainfall of 115mm. The lowest point of the study area has a height of 10m and the highest point is located in Kapsalo at 620m. ” Please provide a reference to support this statement.In Section 3,DEM should be introduced more clearly and detailed.
  2. Was the burned area of second serious fire incident in 2021 close to the limits of the fire that occurred in 2020?
  3. The section of discussion is rather simple.
  4. A section of conclusion should be added. The uncertainties should be addressed.

Round 2

Reviewer 1 Report

Obviously, the authors spent time to work out successfully my comments and they have addressed the concerns that raised in the previous version.

Reviewer 2 Report

I have no further comments.